



# Mitigating impacts of low energy laser pulses on CALIOP data products

Jason L. Tackett[1], Robert A. Ryan[2], Anne E. Garnier[3], Jayanta Kar[3], Brian J. Getzewich[1], Xia Cai[1], Mark A. Vaughan[1], Charles R. Trepte[1], Ron Verhappen[3], David M. Winker[1], Kam-Pui A. Lee[3]

[1]NASA Langley Research Center, Hampton, VA, USA
[2]Coherent Applications, Inc. – Psionic LLC, Hampton, VA, USA
[3]Analytical Mechanics Associates, Hampton, VA, USA

*Correspondence to*: Jason L. Tackett (jason.l.tackett@nasa.gov)

**Abstract.** The spaceborne Cloud-Aerosol Lidar with Orthogonal Polarization (CALIOP) experienced an increasing number
of intermittent low energy laser pulses in the final 7 years of the Cloud-Aerosol Lidar and Infrared Pathfinder Observations (CALIPSO) mission due to coronal arcing within the laser canister as internal pressure levels decreased. Degraded data quality was initially observed primarily over the South Atlantic Anomaly (SAA) region, with impacts eventually spreading globally at lower rates. To preserve the integrity of the CALIOP data record, a suite of low energy mitigation (LEM) procedures was developed to reject data that is substantially impacted by low energy pulses on small, targeted scales to
minimize data loss. LEM corrects level 1B calibration biases of −3 to −4 % and reduces calibration uncertainties by 20–40 % at SAA-latitudes (0–50°S) for the 532 nm daytime and 1064 nm channels. LEM rejection in level 2 processing substantially reduces the occurrence of false feature detections. Horizontally averaged data segments that are LEM-affected (i.e., some low energy shots are present, but LEM determines the filtered profile is still acceptable) experience a signal-to-noise ratio reduction of 6–9 % which increases the probability of false detections, though this is mitigated somewhat by a slight increase
in feature detection thresholds. Features identified in LEM-affected data are similar to unaffected features in terms of measured layer-averaged properties that are important for classification and subtyping. Overall, this evidence suggests that LEM has eliminated the major impacts of low energy pulses on data quality. LEM procedures were implemented in the version 4.51 level 1B data release and are implemented in all data levels of the final version 5 CALIOP data release.

## 1 Introduction

The Cloud-Aerosol Lidar with Orthogonal Polarization (CALIOP) was launched onboard the Cloud-Aerosol Lidar and Infrared Pathfinder Satellite Observations (CALIPSO) satellite in April 2006 as part of a partnership between the United States and French space agencies, NASA and CNES (Winker et al., 2010). CALIOP is a two-wavelength elastic backscatter lidar, operating at 532 and 1064 nm, with polarization-sensitivity at 532 nm. The CALIPSO platform also carried the Imaging Infrared Radiometer (IIR), with three spectral bands optimized for retrieving cirrus optical properties (Garnier et al.,
2012), and a single channel wide-field-of-view camera (WFC) for contextual imagery (Pitts et al., 2007). Over the course of





the 17-year mission, CALIOP delivered unprecedented observations of the vertical profiles of aerosols and clouds, demonstrating the importance of characterizing the vertical structure of Earth's atmosphere. Science data collection was discontinued in June 2023 due to depleted fuel reserves and reduced ability to recharge the spacecraft batteries.

CALIOP consisted of transmitter and receiver subsystems, with the former comprised of two identical lasers with
steerable optics, and the latter comprised of a 1-m telescope, relay optics, and detectors for each of the three channels (Hunt et al., 2009). A payload computer controlled these subsystems, merged measured signals simultaneously acquired in low gain and high gain channels, and averaged data prior to downlink. The transmitter and receiver subsystems performed exceptionally well on average during the mission lifetime, demonstrating the potential longevity of spaceborne lidar systems. However, the greatest data quality challenge experienced by CALIOP was an increasing number of intermittent low energy
laser pulses from the lidar transmitter during the last six years of the mission (mid-2017 to 2023). Many of these low energy pulses, effectively having zero energy, caused biases in the CALIOP 532 nm daytime and 1064 nm calibration coefficients, and degraded level 2 data quality in affected profiles. Figure 1 provides an example of affected data where level 1 attenuated backscatter experienced excessive noisiness and level 2 feature detection results contained a myriad of both false positives and false negatives.

The CALIPSO mission issued a data advisory in late 2018 informing users of the issue and outlined a strategy by which affected data could be avoided. The strategy was simple, but unnecessarily aggressive in the amount of data that it recommended excluding. Because some data loss was inevitable, yet the measurements were so valuable, the mission sought to develop a more deliberate, targeted strategy for identifying and removing affected data across all levels of CALIOP data processing. Collectively, these strategies are termed "low energy mitigation (LEM)". LEM modifications were made to
calibration procedure for the 532 nm daytime and 1064 nm channels in the version 4.51 (V4.51) level 1B data release in 2022. A new version 5 (V5) data release in 2025 includes adjustments to the level 1 energy normalization procedure to account for low energy pulses and a strategic LEM algorithm that alleviates data quality issues in level 2 data products. The goal of this paper is to fully describe the strategies developed to mitigate the influence of low energy pulses on CALIOP observables and characterize data affected by LEM.

The paper is organized as follows. Section 2 describes the origin of low energy laser pulses and characterizes their occurrence frequency during the last six years of the mission. Data used for subsequent analyses is outlined in Sect. 3. Algorithmic details of the LEM algorithm are given in Sect. 4, including its application to level 1B calibration and level 2 data processing. Section 5 characterizes improvements to level 1B calibration coefficients and calibration uncertainties due to LEM, while Sect. 6 describes impacts on level 2 feature detection, atmospheric layer properties, and extinction retrievals.
Section 7 outlines how the CALIOP level 3 data products have adopted LEM into their quality filtering procedures, prior to the summary in Sect. 8.





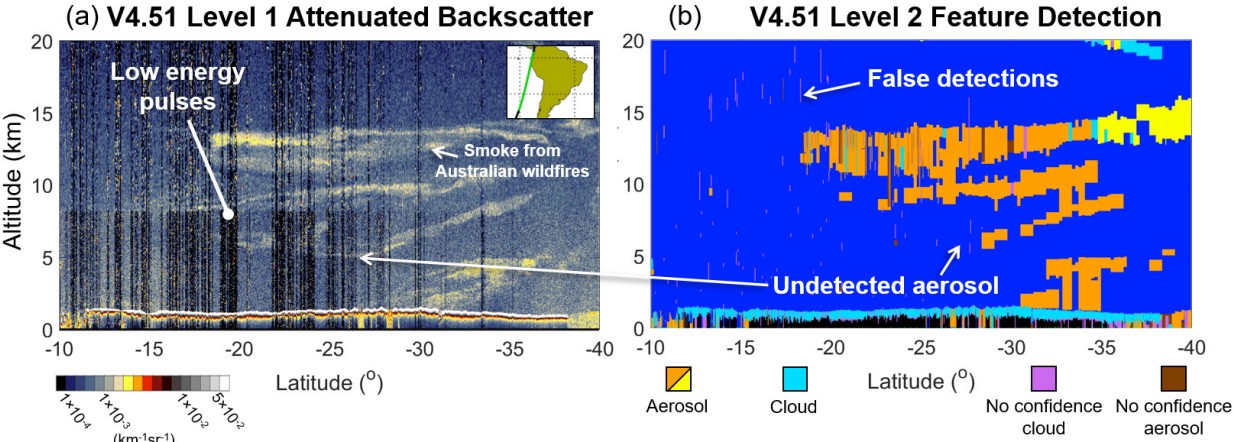

**Figure 1. CALIOP (a) 532 nm attenuated backscatter and (b) level 2 feature detection affected by low energy laser pulses in the V4.51 data product, observed 17 January 2020 at 7:20 UTC.**

## 2 Origin and characteristics of low energy laser pulses

The primary components of the CALIOP transmitter subsystem were two identical Nd:YAG lasers, each housed in a separate pressurized canister (Hunt et al., 2009). The lasers generated light at 1064 nm that was frequency-doubled to generate 532 nm pulses. The laser pulses were formed using an active Q-switch that controlled the optical gain of the laser canister. To monitor performance, sensors were included to monitor laser energy at each wavelength and the pressure within each laser canister. Four years before launch, the canisters were filled with dry air at just above atmospheric pressure and hermetically sealed. Pressure measurements prior to launch indicated that both canisters had slow leaks. It was recognized that canister with the faster leak would eventually reduce the canister pressure to a point where corona discharge could form between the high voltage power supply terminals in the Q-switch and affect its operation. The rate was slow enough, though, that the laser was expected to operate nominally during the planned three-year mission. This laser was designated as the primary laser that would be used in orbit first. The second laser was designated as the backup. Figure 2 displays the pressure trends for the two lasers and the region prone for corona discharges across the Q-switch.



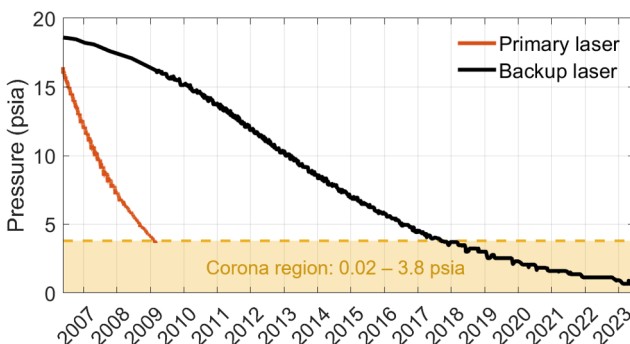

**Figure 2. Canister pressure trends for the primary (red) and backup lasers (black). The predicted corona region is shaded in light orange.**


Over the course of the mission, the majority of energy variations between laser pulses were typically < 1 mJ. Larger energy variations were also observed, but far less frequently, and were partitioned into two classes of anomalous laser

behavior. The first group (Class I) was characterized by energy variations of ~5 mJ and were observed at the beginning of lidar operations. Their cause is unknown but thought to be related to the timing circuitry for the photodiodes that energize the laser slab. The second group (Class II) exhibited more significant energy reductions that ranged from a few millijoules from nominal to complete losses of laser pulses. Their frequency and magnitude of loss increased as the pressure of the laser canister lowered towards pressure levels conducive for corona as defined by Paschen's law (Wadhwa 2007). Their

occurrence was also highly correlated with locations of enhanced solar particle flux such as within the South Atlantic Anomaly (SAA) and the polar regions where the Van Allen radiation belt descends to the altitudes that often affect satellites in low earth orbit (Domingos et al., 2017, Noel et al., 2014). The early occurrence of low energy pulses at pressures above the corona region is attributed to the presence of ionization trails from enhanced energetic particles that can stimulate corona across the Q-switch terminals (Rodriguez et al., 2022).

Beginning in December 2008 and into February 2009, several bursts of near-zero energy pulses occurred that impacted laser operations as the pressure in the canister approached the corona threshold. Because of likelihood for additional bursts, a decision was made to switch to the backup laser in March 2009. The backup laser operated nearly flawless until the first signs of Class II low energy pulses were observed in mid-2016 over the SAA with growing frequency. Class I low energy pulses were also observed after this time, but their presence was often masked by the more frequent Class

II low energy pulses. Because of the added understanding on the likely causes of low energy pulses gained from extended performance characterization studies conducted with the backup laser, changes were made to the payload's operating procedures that greatly reduced the impact by bursts of low energy pulses during routine lidar operations. In this manner, the backup laser was able to continue operations at pressures far below the Paschen threshold.

The distributions of 532 nm laser energies inside the SAA are shown in Fig. 3 for time periods before and after the

onset of low energy pulses for the backup laser. Similar behavior is noted for the 1064 nm channel. A subset of affected



pulses has energies reduced by less than 20 mJ from the nominal 95 mJ value and another subset has energies that increased by 5–10 mJ. Energy deviations of ±10–20 mJ from nominal are easily accommodated by the energy normalization process during level 1A processing (Sect. 4) and have no notable noticeable impact on data quality other than a minor reduction in the signal-to-noise ratio (SNR). The more detrimental subset of pulses are those Class II anomalies with energies < 10 mJ,
which effectively indicates zero energy from the laser. Non-zero values occurred because the laser energy monitors do not report values of exactly zero when there is no energy incident upon them. There is an offset in the conversion from digital data to engineering units, yielding a minimum reported energy around 4 mJ for the 532 nm channel. Likewise, the 1064 nm energy monitor has a minimum reported energy of ~40 mJ. For the remainder of this paper the term "low energy" will refer only to laser pulses with near-zero energy since these are the cause of data quality issues that must be mitigated. Energy in
the 532 nm channel will be used to identify low energy pulses because (1) they also identify near-zero energy at 1064 nm, and (2) the 1064 nm channel is calibrated relative to the 532 nm channel.

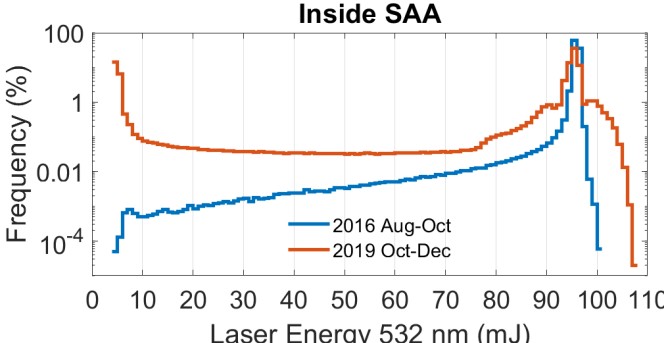

**Figure 3. Distribution of 532 nm laser energy pulses within the SAA before the low energy issue (blue; August–October 2016) and**
**during (red; August 2021).**

During the latter years of the mission, the occurrence of low energy pulses (532 nm energy < 10 mJ) expanded outside the SAA (Fig. 4) and increased in frequency (Fig. 5). Though an increase in frequency of low energy pulses was expected as the laser canister pressure fell, the non-monotonic behavior of the frequency time series in Fig. 5 is likely a
result of several factors that are not entirely understood. By 2017, the monthly occurrence frequency of low energy pulses exceeded that experienced by the primary laser which had maxima < 0.3% inside the SAA and < 0.004% outside the SAA. During 2017 to early-2019, the frequency of low energy pulses increased slowly, primarily inside the SAA before an acceleration in mid-2019 through early 2020 which culminated in frequencies of 30% inside the SAA and 1.6% outside. A sudden decrease in frequency occurred in March 2020, followed by a period of somewhat stable low energy pulse
frequencies from mid-2020 through October 2021. Another increase occurred during late 2021 to early 2022, reaching frequencies of 55% inside the SAA and highly variable monthly frequencies outside the SAA for the majority of 2022. An unexpected decline in frequency of low energy pulses occurred in the last six months of the mission. For several orbits



during the spring of 2023, CALIOP experienced nearly constant low energy pulses through most of the dayside (e.g., Fig. S1), causing the two vertical bands of enhanced frequency in Fig 4.


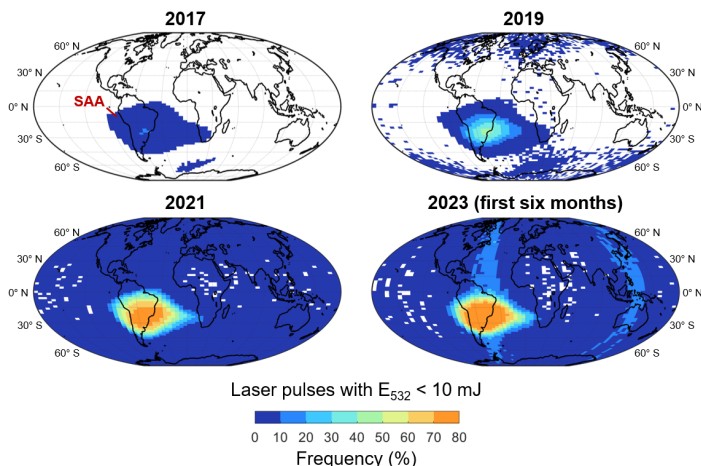

**Figure 4. Frequency of laser pulses with 532 nm energy < 10 mJ during 2017, 2019, 2021, and the first six months of 2023. Vertical bands of enhanced frequencies in the 2023 map are due to several orbits of nearly constant low energy pulses.**

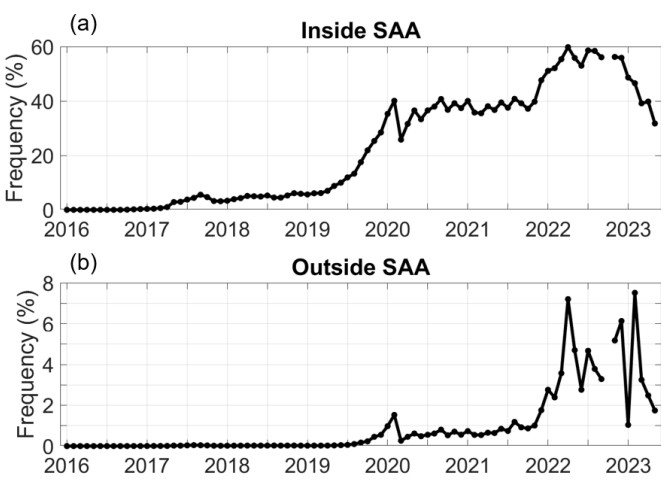

**Figure 5. Frequency of laser pulses with 532 nm energy < 10 mJ (a) inside and (b) outside the SAA.**


The number of consecutive low energy pulses also evolved during the 2017–2023 timeframe. Whereas single, isolated low energy pulses were most common in 2018, accounting for 86% of all low energy pulses inside the SAA and 99% outside the SAA, occurrences of multiple, consecutive low energy pulses became more common in later years. Figure S2 shows the frequency of consecutive low energy pulses for April 2018, 2021, and 2023. In early 2021, 90% of low energy pulses occurred in sequences of < 8 consecutive shots inside the SAA and < 4 consecutive shots outside the SAA. By early




2023, this had increased such that 90% of low energy pulses occurred in sequences of < 12 consecutive shots in the SAA and < 10 consecutive shots outside the SAA. Clearly the behavior of low energy pulses had changed with time.

The mechanisms responsible for the observed variability in the frequency of low energy pulses may never be fully understood. Nonetheless, the data quality of affected profiles was inarguably degraded, particularly over South America and the tropical Atlantic Ocean. It is the goal of the LEM algorithms to mitigate the impacts of low energy pulses and to restore the integrity of the CALIOP data record.

## 3 Data used

CALIOP level 1B and level 2 data products are used for the analysis in this paper. The CALIOP level 1B data product contains calibrated, geolocated profiles of attenuated backscatter for the three CALIOP measurement channels. Calibration methods are described in Sect. 5. Level 2 layer products summarize properties of detected particulate layers while level 2 profile products provide retrieved profiles of extinction coefficients among other parameters within the detected layers. Atmospheric layers are detected in CALIOP processing using an iterative, multi-resolution feature detection algorithm

(Vaughan et al., 2009) and then classified as aerosols or clouds by comparing measured layer properties against multi-dimensional probability density functions for known feature types (Liu et al., 2019). Cloud ice-water phase is determined based on measurements of depolarization and layer-integrated attenuated backscatter (Hu et al., 2009; Avery et al., 2020). Vertically resolved profiles of particulate backscatter and extinction coefficients are retrieved for all feature types using the family of algorithms described in Young et al. (2018).

CALIOP version 5.0 (V5) level 1B and level 2 data products contain all the low energy mitigation modifications discussed in the next section. Version 5 level 1B data are compared to previous releases including V4.51 (released in 2022) and V4.1 (released in 2016). Version 5 level 2 merged layer products are evaluated to demonstrate the impacts of low energy mitigation (LEM) modifications. Additional data used for verification are the CALIPSO IIR 12.05 μm brightness temperature differences co-located with CALIOP and reported at 1 km resolution in the IIR V4.51 level 2 track product.

These brightness temperature differences ($BTD_{oc}$) represent the difference between IIR measured observations at 12.05 μm and calculations derived from 12.05 μm simulations in cloud-free conditions (Garnier et al., 2021). A polygon of latitude-longitude boundaries is also used to identify profiles within the SAA region (Appendix A).

## 4 Algorithm changes to mitigate low laser energy impacts

The overarching goal of low energy mitigation (LEM) is for CALIOP data affected by low energy laser pulses to have

quality similar to that of unaffected data in terms of calibration accuracy, signal-to-noise ratio (SNR), and feature detection reliability. Data loss due to excessive numbers of low energy pulses is minimized by rejecting data in small, targeted segments. To understand how these segments are selected, Sect. 4.1 provides a brief overview of how CALIOP



measurements acquired onboard the spacecraft are averaged prior to downlink and subsequently re-gridded to generate the level 1B data product. Adjustments to the energy-normalization procedure to account for LEM are also introduced. Details of the LEM algorithm is given in Sect. 4.2.

### 4.1 CALIOP onboard averaging and level 1A to level 1B processing

The CALIOP lidar return signals are measured in units of digitizer counts at a spatial resolution of 333 m horizontal by 15 m vertical (Hunt et al., 2009). To reduce downlinked data volume, the measured signals are averaged onboard by the payload computer to different resolutions commensurate with the spatial variability expected for atmospheric features within different altitude regimes. Downlinked data are organized into "frames" having a horizontal extent of 5 km (15 consecutive laser pulses). The schematic in Fig. 6 shows how data are averaged within each 5 km frame for the 532 nm channel. Altitude regions are numbered 1–5, and each has varying amounts of horizontal and vertical averaging. Note that the primary concern of LEM is horizontal averaging, so vertical averaging will no longer be discussed beyond noting that there are different numbers of range bins within each altitude region and a total of 583 range bins are downlinked per frame.

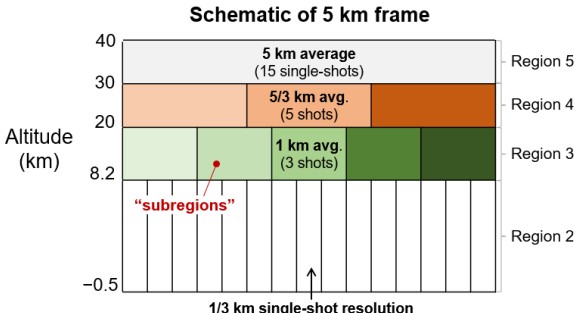

**Figure 6. Schematic of a 5 km frame, identifying altitude regions and onboard averaging scales. Altitude region 1 is excluded for clarity (−2 to −0.5 km). It has identical horizontal resolution as region 2.**

| Region | Altitude (km) | Vertical resolution per subregion (m) | Horizontal resolution per subregion (km) | N single shots averaged per subregion ($N_{avg}$) | N subregions per 5 km frame |
|---|---|---|---|---|---|
| 5 | 30.1 – 40.0 | 300 | 5 | 15 | 1 |
| 4 | 20.2 – 30.1 | 180 | 5/3 | 5 | 3 |
| 3 | 8.2 – 20.2 | 60 | 1 | 3 | 5 |
| 2 | −0.5 – 8.2 | 30 | 1/3 | 1 | 15 |
| 1 | −2.0 – −0.5 | 300 | 1/3 | 1 | 15 |

**Table 1. Definitions for altitude regions, subregions, and resolutions of on-board averaging at 532 nm.**





Below 8.2 km (regions 1 and 2), data is downlinked at full 333 m horizontal resolution, called "single-shot resolution". Above 8.2 km, data in each frame is horizontally averaged within several "subregions", where the number of subregions depends on the altitude region. For example, in region 3 there are five subregions, each an average of three single shots, yielding 1 km resolution in the horizontal dimension for each range bin. Crucially, only one averaged value is downlinked for each range bin within a given subregion. The consequence is that a single low energy pulse within a subregion can irrevocably degrade the data quality of the subregion even if the other pulses have nominal energy because their signals have been averaged on board the satellite.

After downlink, vertically continuous profiles are generated in level 1A processing by first establishing a grid for each 5 km frame having single-shot horizontal resolution for all five altitude regions. Single-shot resolution profiles in regions 1 and 2 are mapped directly onto this grid. By contrast, the downlinked signal profiles for regions 3, 4, and 5 are replicated as necessary to span the horizontal resolutions for each subregion. Continuing the previous example, the profile downlinked at 1 km horizontal resolution in a region 3 subregion is replicated horizontally across all three single-shot resolution grid cells in the subregion. This is called "pseudo-single-shot" resolution because the profiles are reported at single shot resolution but are not truly single shot resolution measurements.

After gridding, the signals are normalized by energy and gain. In regions 1 and 2, energy and gain normalization is described by Eq. (1) where $S$, $E$, and $G_{\mathrm{A}}$ are the downlinked signal in counts, the laser energy, and the amplifier gain at wavelength $\lambda$, respectively. The variables $i$ and $z$ denote, respectively, the gridded profile index and range bin.

$$S_{\mathrm{norm},\lambda}(i,z) = \frac{S_\lambda(i,z)}{E_\lambda(i)G_{\mathrm{A},\lambda}} \tag{1}$$

In the pseudo-single-shot altitude regions, the mean downlinked signal $\overline{S}_\lambda(z)$ is normalized by the mean laser energy $\overline{E}_\lambda$ and mean gain $\overline{G}_{\mathrm{A},\lambda}$ over each subregion. This was defined in the V4.51 level 1 data product and earlier versions by:

$$\overline{S}_{\mathrm{norm},\lambda}(i,z) = \frac{\overline{S}_\lambda(i,z)}{\overline{E}_\lambda(i)\overline{G}_{\mathrm{A},\lambda}} \tag{2}$$

An adjustment has been made to the energy normalization procedure in the pseudo-single-shot regions with the V5 level 1B data release to account for low energy pulses. This adjustment was necessary because near-zero energy pulses bias the normalized signal low in Eq. (2) while preserving background noise. To restore the signal magnitude, the mean laser energy is now computed for each subregion based on the energy levels of profiles with "good" energy, yielding $\overline{E}'_\lambda$ (Eq. 3).





These are profiles where the 532 nm energy is greater than 80 mJ, a threshold selected to ensure only values near the nominal energy peak are used. In subregions without good energy pulses, $\bar{E}'_\lambda$ is set to 1 mJ. The new equations for energy and gain normalization in V5 are below, where $N_{\text{avg}}$ is the number of single-shot profiles in the subregion, $j$ is the profile index within the subregion, and $N_{\text{good}}$ is the number of profiles with nominal laser energy.

$$\bar{E}'_\lambda = \frac{1}{N_{\text{good}}} \sum_{j=1}^{N_{\text{avg}}} E_\lambda(j), \quad \text{if } E_{532}(j) > 80 \text{ mJ} \tag{3}$$

$$N_{\text{good}} = \sum_{j=1}^{N_{\text{avg}}} 1, \quad \text{if } E_{532}(j) > 80 \text{ mJ} \tag{4}$$

$$\overline{S}_{\text{norm},\lambda}(i,z) = \frac{\overline{S}_\lambda(z)}{\bar{E}'_\lambda G_{A,\lambda}} \tag{5}$$


The 532 nm attenuated backscatter ($\beta'_{532}$) comparison in Fig. 7 demonstrates the revised energy normalization procedure improves SNR in regions 3 and higher (> 8.2 km) for profiles affected by low energy pulses. In this example, SNR increases by 50 % at 15–20 km and by 10 % at 20.1–25 km over latitudes 5°–35° south. SNR is computed as the mean divided by standard deviation of $\beta'_{532}$ multiplied by the square root of number of single-shot samples within the sample

volume (i.e., SNR $= \sqrt{N} \cdot \mu/\sigma$ ). There is also a decrease in the frequency of false feature detections in this region, discussed further in Sect. 6. As expected, SNR is still poor for these profiles in the single-shot resolution altitude regions below 8.2 km.



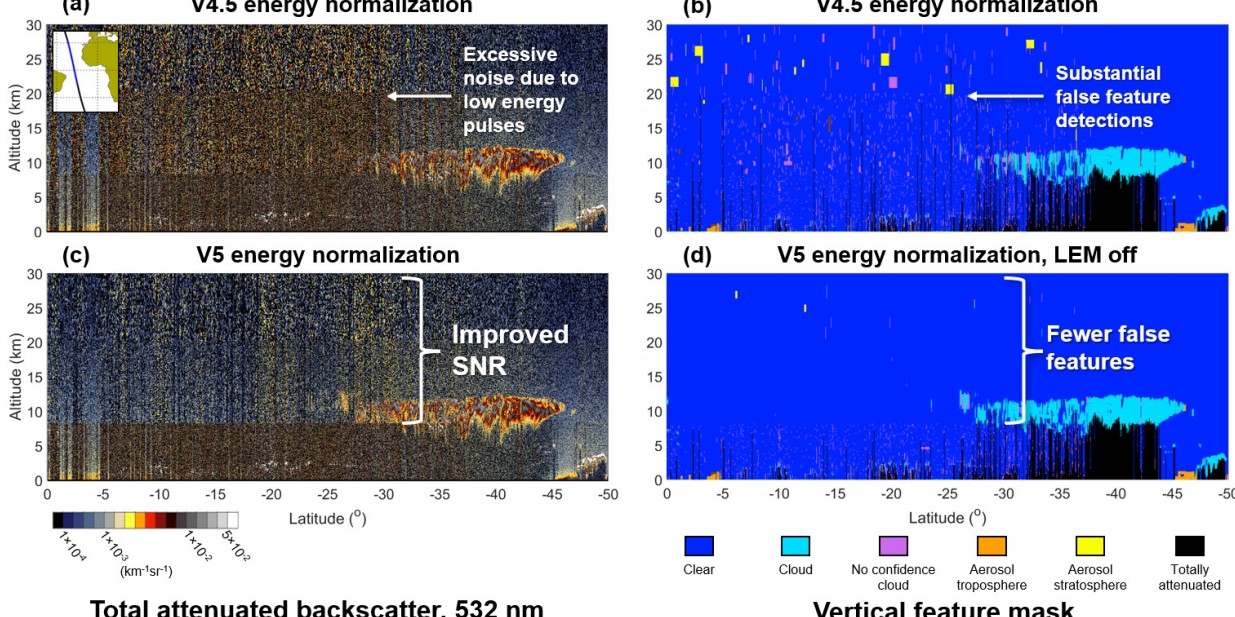

**Figure 7. Panels (a) and (b) show, respectively, the level 1 532 nm total attenuated backscatter and level 2 vertical feature mask**
**from the V4.51 release for a granule strongly affected by low energy pulses on 02 June 2021 at 16Z. Panel (c) shows that the V5**
**532 nm total attenuated backscatter for the same granule exhibits a noticeable increase in SNR above 8.2 km relative to panel (a)**
**due to the adjusted energy normalization procedure. Improvements to layer detection attributed solely to the adjusted energy**
**normalization are illustrated in panel (d), which shows the vertical feature mask generated from data in panel (c) but without**
**applying the V5 level 2 LEM algorithm (Sect. 4.2).**


Following energy and gain normalization, profiles are geolocated and altitude-registered, then replicated as necessary to

create the pseudo-single shot resolution data reported in the level 1B product (Table 1). The final step is range correction

$(r^2)$ and application of the calibration coefficient, $C$, yielding attenuated backscatter $\beta'$, as given by Eq. (6) (Kar et al.,

2018). This quantity, reported in the level 1B data product, provides the raw material used by the level 2 LEM algorithm.

Sect. 4.3 will detail how LEM is applied to CALIOP calibration procedures.

$$\beta'_\lambda(i,z) = \frac{r^2(i,z)}{C_\lambda(i)} \overline{S}_{\text{norm},\lambda}(i,z) \tag{6}$$

**4.2 Low energy mitigation (LEM)**

Profiles containing low energy laser pulses are identified as those having 532 nm energy less than a minimum energy

threshold, $E_{\text{min}}$, as in Eq. (7). Different values are used depending on the application. Energy normalization uses 80 mJ as the

threshold, the 532 nm daytime and 1064 nm calibration procedures use 10 mJ, and level 2 LEM uses 50 mJ. In practice, all





three thresholds yield similar results since most energies tend to be very low (< 10 mJ) or nominal (> 80 mJ), with few values in between (Fig. 3).

$$E_{\text{low}} = E_{532} < E_{\min} \qquad\qquad (7)$$

Segments of data having low energy (i.e., frames or subregions) may or may not be rejected depending on the requirements outlined in this section. Data that are not rejected are considered "accepted" by LEM. It is important to note that rejection due to LEM only occurs when computing level 1B calibration coefficients and during the generation of level 2 data products. Importantly, LEM rejection is not applied to the contents of the level 1B data product. For data to be used for calibration or level 2 processing, it must pass LEM acceptance requirements, which break down into three categories. Table

2 lists the specific requirements for each category.

| Subregion acceptance requirements |
| --- |
| Region 1 and 2 profiles with low energy are rejected |
| Region 3 subregions must have at least 2 (out of 3) profiles with $E_{532} > E_{\min}$ |
| Region 4 subregions must have at least 2 (out of 5) profiles with $E_{532} > E_{\min}$ |
| *Some* unrejected data must exist in each 1 km segment of regions 2 and 3 |
| **Frame acceptance requirements** |
| Region 2 must have at least 6 (out of 15) accepted profiles |
| Region 3 must have at least 3 (out of 5) accepted subregions |
| Region 4 must have at least 1 (out of 3) accepted subregions |
| **Coarse resolution feature detection requirement** |
| At least 75 % of frames must be accepted at 20 km resolution (3 out of 4) |
| At least 75 % of frames must be accepted at 80 km resolution (12 out of 16) |

**Table 2. LEM acceptance requirements for subregions, frames, and coarse resolution feature detection.**

Subregion acceptance requirements dictate the number of low energy pulses that can be tolerated before a subregion

must be rejected. This allows targeted data rejection on the smallest spatial scale possible. In the single-shot resolution regions 1 and 2, range bins having low energy are simply rejected. Subregions in higher altitude regions can include a limited number of low energy pulses in their averages because energy normalization compensates for the reduced mean energy. The requirement for 2 out of 3 profiles with good energy in region 3 subregions (at 8.2 – 20.1 km) ensures at least 86% of the nominal SNR is achieved at altitudes where cloud layers are common (SNR $\propto \sqrt{N_{\text{good}}}$). This requirement is

relaxed somewhat for the higher altitude region 4 subregions because fewer features typically exist there. The final subregion





acceptance requirement in Table 2 ensures that a continuous $\beta'$ profile exists at 1 km resolution between regions 2 and 3 after all other subregion acceptance requirements have been applied. This is critical for level 2 planetary boundary layer (PBL) cloud-clearing which begins its scan at 1 km resolution in search of single-shot resolution clouds (Vaughan et al., 2009).

Frame acceptance requirements govern the number of rejected subregions or profiles that can be tolerated before the entire 5 km frame is rejected. Rejecting full frames avoids unreliable feature detections and extinction retrieval errors that would occur as SNR decreases due to the reduction in contributing samples. Since 5 km is the fundamental horizontal averaging resolution for CALIOP's spatial and optical properties retrievals, rejecting entire 5 km frames when too much data have been rejected at the subregion scale minimizes perturbations to the performance of these level 2 algorithms. The

amount of accepted data required ensures that a sizeable portion of the SNR expected for a 5 km average is achieved. For example, frames meeting the frame acceptance requirements in Table 2 will achieve > 63% of expected SNR in region 2 and > 77% in region 3. Note that subregion and frame acceptance requirements are tightly coupled such that multiple requirements can simultaneously cause frame rejection depending on the ordering of low energy pulses within the 15-shot frame. For this reason, frames are more often rejected before the number of accepted profiles declines to 6; they are more

likely to have 7 or more accepted profiles, yielding > 68% of expected SNR in region 2.

Coarser resolution acceptance requirements are specific to level 2 feature detection at horizontal resolutions of 20 km and 80 km. These requirements specify how many frames can be rejected before feature detection is not performed at the coarser resolution. Following PBL cloud clearing, the level 2 algorithm searches for features at 5 km horizontal resolution, removes all that are detected, and then re-averages the data remaining in the four intensity-cleared 5 km averages to form a

single profile with a nominal horizontal resolution of 20 km. The search continues for features at 20 km resolution; all features found are removed, and the data remaining in the four 20 km resolution profiles are averaged into a single profile with a nominal 80 km resolution that is used for the final search for features at 80 km horizontal resolution (Vaughan et al., 2009). Features detected at 20 and 80 km horizontal resolution can have relatively low SNR, as features with higher SNR may have been detected and removed in the 5 km profile scans. In order to prevent retrieval errors caused by lower than

typical SNR, LEM requires at least 75% of frames to be accepted in the 20 or 80 km average for feature detection to be attempted at these resolutions.

## 4.3 How LEM-rejected data appears in level 2 data products

Several flags indicate where and why data is rejected by LEM in level 2 data products. A new flag value of −111 is assigned to LEM-rejected data within Science Data Sets (SDSs) containing retrieved and layer-aggregated quantities, excluding bit-

mapped SDSs. Three new quality control (QC) flags have also been added. The Low Energy Mitigation Column QC Flag contains bit-mapped values for each profile indicating if it has LEM-affected data; which subregion, frame, and coarse resolution feature detection requirements were violated; and which altitude regions contain LEM-rejected data (Table B1). The vertical feature mask (VFM) product includes a new VFM Feature Detection Quality Flag which contains bits



identifying which range bins contain low energy pulses; which have been rejected by LEM; and what resolutions were used
for feature detection (Table B2). A Boolean flag has also been added to indicate when layers are affected by LEM: the Low
Energy Mitigation Feature QC Flag (Table B3). When this flag is true, the corresponding atmospheric layer is either partially
missing samples due to LEM rejection or it contains low energy pulses in pseudo-single-shot subregions that LEM does not
reject. Lastly, the feature type field in the feature classification flags is assigned as invalid in LEM-rejected segments of level
2 data products rather than cloud, aerosol, etc.

The suite of LEM QC flags provide transparency into the level 2 LEM process and give data users options for
additional quality filtering if desired. Section 6 will demonstrate that the quality of level 2 data affected by low energy pulses
(but not rejected) similar to unaffected data when LEM is implemented, so for most applications there should be no need for
additional filtering. However, if a data user would like to entirely avoid data impacted by LEM, the Low Energy Mitigation
Column QC flag should be consulted to ensure no bits are set. This same flag could be used to exclude profiles where feature
detection was not attempted at 20 or 80 km horizontal resolution by evaluating bits 4 and 5. Considering when feature
detection is not operating at its nominal resolutions can be important for interpreting long-term aerosol and cloud occurrence
trends based on CALIOP detection because weakly backscattering layers are not fully represented. As a final example, users
wishing to analyze level 1B data can use the single-shot resolution version of this flag to reject low energy affected data in
the same manner as LEM. Note that because of the continuity between regions requirements, using this flag yields different
results than simply checking the laser energy.

## 4.4 Application of LEM to level 1B calibration

Starting with the level 1 V4.51 release in 2022, the LEM algorithm described in Sect. 4.2 was implemented in the 532 nm
daytime calibration process and a more restrictive version of LEM was applied for the 1064 nm calibrations. The
fundamental calibration for CALIOP is the 532 nm nighttime channel, which uses a molecular normalization technique that
computes calibration coefficients as a function of granule elapsed time from the terminator (Kar et al., 2018). Calibration
coefficients are computed at 36–39 km as the ratio of the range-corrected, gain and energy-normalized signal to the
attenuated backscatter expected for a pure molecular atmosphere based on MERRA-2 reanalysis data. To achieve sufficient
SNR, CALIOP signals are harvested and averaged over 55 km along-track and ±5 granules across-track segments (i.e., across
adjacent granules matched by granule elapsed time). The signal-harvesting process includes an adaptive spike filter that
excludes signals falling outside of minimum and maximum thresholds based on the expected random noise within the
profile, and a seasonally and latitudinally varying noise-to-signal ratio (NSR) threshold (Kar et al., 2018). These filters
effectively remove data affected by low energy pulses due to their low SNR. For this reason, the 532 nm nighttime
calibration coefficients are unaffected by the low energy issue.

        Daytime 532 nm calibration coefficients were affected by low energy pulses prior to the V4.51 data release,
however. The 532 nm daytime calibration technique transfers the 532 nm nighttime calibration to daytime by comparing
uncalibrated 532 nm daytime cloud-free scattering ratios (the ratio of molecular + particulate backscatter to molecular



backscatter) to calibrated 532 nm nighttime scattering ratios as a function of granule elapsed time (Getzewich et al., 2018). The transfer occurs at altitudes above the 400 K isentropic surface (typically 17–25 km, depending on latitude), where aerosol loading is assumed to be diurnally invariant. Far more substantial averaging is required for adequate daytime SNR compared to 532 nm night: scattering ratios are based on averages computed 200 km along-track and over ±52 across-track granules (> 7 days). In earlier data releases, the large amount of across-track averaging allowed low energy pulses in affected granules to contaminate the calibration coefficients applied to adjacent granules. The LEM algorithm was implemented in the V5 532 nm daytime calibration process to reject data containing low energy pulses during scattering ratio harvesting using an $E_{min}$ threshold of 10 mJ. This threshold is defined to exclude the peak of near-zero energy pulses (Fig. 3).

Likewise, 1064 nm calibration coefficients were affected by low energy pulses prior to the V4.51 release. The 1064 nm calibration strategy also transfers the 532 nm calibration coefficients to 1064 nm, but instead uses an ensemble of "calibration quality ice clouds" as a reference, assuming the 1064 nm to 532 nm backscatter coefficient color ratio equals 1.01 (Vaughan et al., 2019). Substantial averaging of calibration quality cloud signals is required to compute "scale factors" that transfer the calibration: the nominal averaging window is ~605 km along-track and ±54 adjacent granules across-track. To avoid low energy pulses from biasing the harvested scale factors, a restrictive LEM quality filter was implemented in V4.51 that disqualifies ice clouds as being calibration-quality if they contain any low energy pulses. An $E_{min}$ threshold of 10 mJ was also used.

**5 LEM improvements to level 1B calibration**

The impact of low energy pulses on the V4.1 daytime calibration process is demonstrated in Fig. 8 (red lines). During November 2020, a low bias of −3 % existed in the 532 nm daytime calibration coefficients at SAA latitudes due to scattering ratios affected by low energy pulses from within the SAA contributing to the ~7 day across-track average (Fig. 8a). The low bias was propagated into the 1064 nm calibration coefficients through calibration quality ice cloud signals identified during the 532 nm to 1064 nm calibration transfer process, yielding a −4 % low bias in the 1064 nm calibrations at the same latitudes (Fig. 8b). Because some samples within the SAA that contributed to the along-track average were devoid of signal, contributing only noise, the calibration coefficient relative uncertainties became exceptionally large, reaching 20–40 % in the zonal mean (Fig. 8c-d). The implementation of LEM procedures in V4.51 corrected these calibration biases and lowered their relative uncertainties at SAA latitudes to 2–3 % (black lines). This is still higher than relative uncertainties at mid-latitudes (~1 %) due to the smaller number of contributing samples after LEM rejection. There are no changes in the 532 nm nighttime calibration coefficients since the spike and NSR filters used in the 532 nm nighttime calibration process already removes low energy pulses. Changes in the 1064 nm nighttime calibration coefficients due to LEM rejection of calibration quality ice cloud candidates are small on average (~1–2%). Minor changes outside the SAA are due to polarization gain ratio adjustments in V4.5 that are unrelated to laser energy issues (Vaughan et al., 2023).




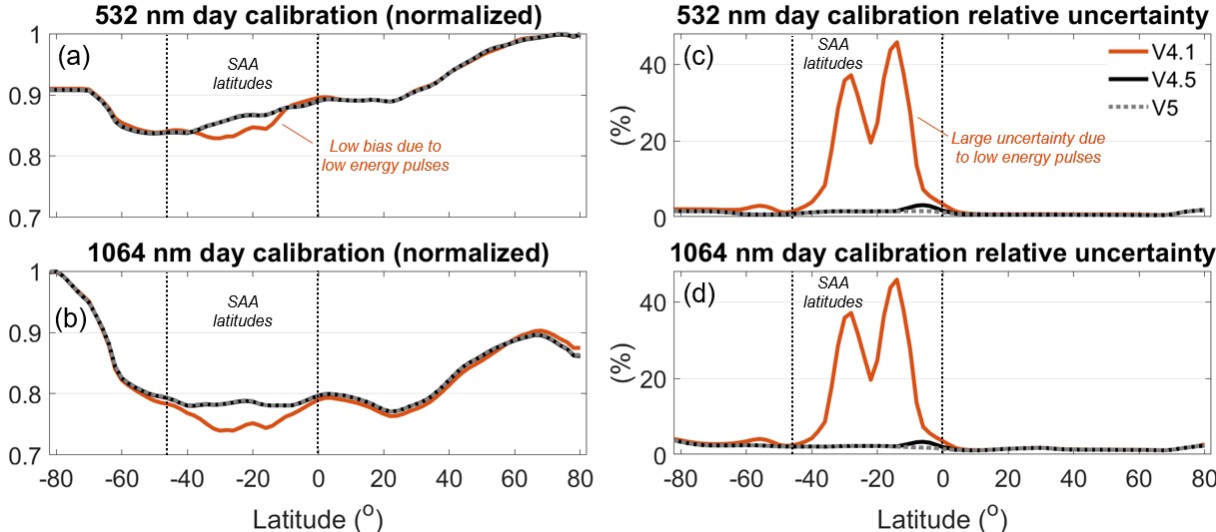

**Figure 8. Zonal mean daytime (a, b) calibration coefficients and (c, d) calibration relative uncertainties for the 532 nm and 1064 nm channels in the V4.1, V4.51, and V5 data releases for November 2020. Calibration coefficients are normalized by the maximum of all three versions.**

## 6. LEM impacts on level 2 retrievals

Profiles containing low energy pulses in V5 level 2 products will either be fully rejected by LEM or they will be accepted, provided that LEM acceptance requirements are fulfilled. Those that contain low energy pulses and are accepted are called "LEM-affected profiles". Data within LEM-affected 5 km resolution profiles (i.e., frames) may have marginal reductions in SNR either due to intra-frame rejection, or by allowing a tolerable number of low energy pulses to remain in the average. The time series in Fig. 9 shows how the frequency of rejected and affected 5 km frames evolve during the last eight years of the mission. Whereas 40 % of frames are rejected inside the SAA between 2020–2022, only ~0.5 % are rejected outside the SAA during the same timeframe. LEM-affected frames occur outside the SAA at a rate of 1–3 %. The final 1.5 years of the mission experienced higher frame rejection rates inside the SAA, with decreasing occurrences of LEM-affected frames as total data rejection became more probable in that region. Outside the SAA, frame rejection rates increased to 3–8 % while LEM-affected rates remained around 3 %. Feature detection at 20 and/or 80 km resolution is disallowed in a subset of LEM-affected frames at rates similar to that of rejected frames in Fig. 9(b) due to LEM coarse resolution feature detection requirements (Table 2).



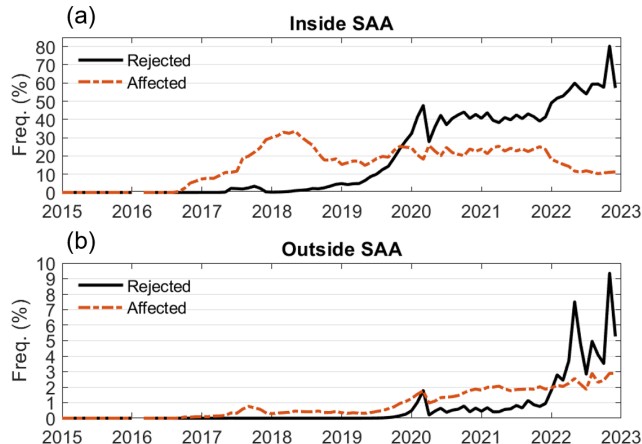

**Figure 9. Frequency of level 2 LEM-rejected and LEM-affected 5 km frames (a) inside the SAA and (b) outside the SAA.**

**6.1 Impacts on feature detection**

The primary impact of LEM on level 2 data products is the near elimination of false feature detections with some recovery of undetected features. As demonstrated in Fig. 7(d), the correction to energy normalization substantially reduces their frequency in profiles affected by low energy pulses for altitudes above 8.2 km, yet lower altitudes are still prone to frequent false detections prior to enabling LEM in level 2 processing. Figure 10 shows the improvement after LEM is enabled on the vertical feature mask for this case: profiles containing false detections below 8.2 km have been rejected and false features at

higher altitudes have been minimized. In addition, previously undetected aerosol is now detected below 1 km between 0–15° S (inset).

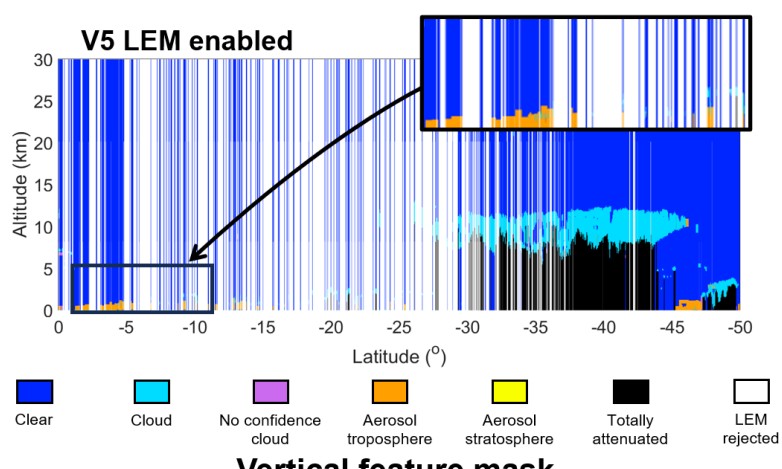

**Figure 10. V5 level 2 vertical feature mask with LEM enabled for the case in Fig. 7.**





The cloud-aerosol discrimination (CAD) score provides a useful metric to characterize the efficacy of false feature rejection by LEM. It is computed for each feature by the CAD algorithm using layer-mean attenuated backscatter at 532 nm $\langle\beta'_{532}\rangle$, attenuated backscatter color ratio $\chi' = \langle\beta'_{1064}\rangle/\langle\beta'_{532}\rangle$, volume depolarization ratio $\delta_v = \langle\beta'_\perp\rangle/\langle\beta'_\parallel\rangle$, and altitude/latitude information (Liu et al., 2019). Valid CAD scores range from −100 to +100, where aerosol layers are negative and cloud layers are positive. Feature classification confidence is highest for CAD score magnitudes closest to 100, which indicates the measured layer properties are most likely representative of the assigned feature type (cloud or aerosol). There is no confidence in feature classification for layers with CAD scores near zero. Special CAD scores, occurring infrequently, are assigned due to anomalous layer properties (and assigned no confidence) or in situations where the feature type is specified a priori rather than calculated by the CAD algorithm (assigned high confidence).

The key characteristics of low-energy-generated false features are unusually low color ratios, resulting in cloud classifications with CAD scores < 10, typically near zero. Figure 11 illustrates the anomalous layer properties of the false features reported in V4.51 for the Fig. 7(d) case prior to LEM implementation. Typical cloud and aerosol populations are also noted.

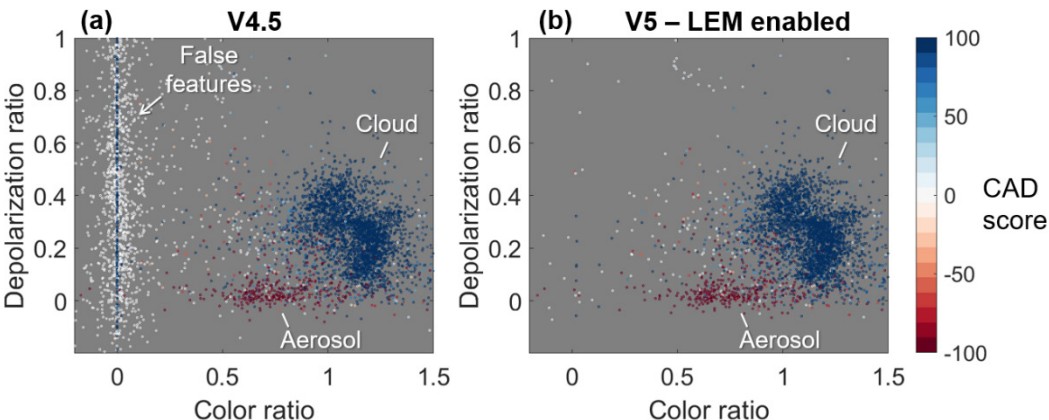

**Figure 11. Depolarization ratio and attenuated backscatter color ratio for all 5 km and coarser resolution features detected in (a) V4.51 and (b) V5 with LEM enabled for the case in Fig. 7(d).**

To evaluate how well LEM targets these features, two test datasets were generated from the V5 level 2 codebase: one with LEM enabled and another with LEM disabled. Table 3 summarizes the global comparison for the year 2021. Most layers exist in both test sets (87 %) and are considered unaffected by LEM (top row). This represents typical feature classification with nominal data, where cloud classifications are more common than aerosol, and high confidence classifications are more common than no-confidence classifications. Around 9% of layers are rejected when LEM is enabled, and most were classified as no-confidence clouds when LEM was disabled (65 %), as expected (middle row). The impact of their rejection is also evident in Fig. 11(b). Some cloud layers with high classification confidence are also rejected (22 %),




though their frequency is small relative to the total number of clouds detected (2.5 % of all clouds). New features occur at a low rate (3.6 %), typically having high classification confidence (bottom row).

| | Proportion of all layers | No confidence cloud | No confidence aerosol | High confidence cloud | High confidence aerosol |
|---|---|---|---|---|---|
| Not affected by LEM | 87.0 % | 6.6 % | 1.4 % | 68.7 % | 16.3 % |
| Rejected by LEM | 9.4 % | 65.0 % | 2.5 % | 22.0 % | 4.9 % |
| New with LEM | 3.6 % | 9.1 % | 2.3 % | 59.3 % | 19.4 % |

**Table 3. Global distribution of feature classification confidence for layers that are not affected, rejected, and newly detected due to LEM in the year 2021. The first column is the proportion within each of these categories out of all layers. The final four columns are the proportion of layers within each category having the indicated confidence level. High confidence layers: $70 \leq |CAD\ score| \leq 100$; no confidence layers: $0 \leq CAD\ score \leq 20$. The feature type QA bits in the feature classification flag are queried for layers with special CAD scores to ascertain confidence level.**

LEM also rejects profiles where features truly exist yet are undetected by the level 2 feature finder due to excessive low energy pulses. These profiles are incorrectly classified as cloud-free without LEM enabled. This is illustrated by Fig. 12, which shows CALIPSO's IIR 12.05 μm $BTD_{oc}$ over ocean for IIR pixels co-located with these 5 km resolution profiles. In this analysis, so called "cloud-free" profiles do not have clouds reported by CALIOP at single-shot, 5 km or 20 km horizontal resolutions and each 1 km resolution IIR observation does not contain LEM-rejected data (relevant only when LEM is enabled). For the LEM-disabled case (blue), 30 % of pixels have $BTD_{oc}$ less than −3 K, which unambiguously indicates the presence of undetected clouds and misrepresented cloud cover. Ordinarily, the $BTD_{oc}$ distribution is centered at near-zero with a standard deviation of less than 1 K for cloud-free profiles (Garnier et al., 2021), as in the LEM-enabled cases (red, yellow). Note that no significant difference is seen between IIR pixels belonging to LEM-unaffected and LEM-affected profiles.



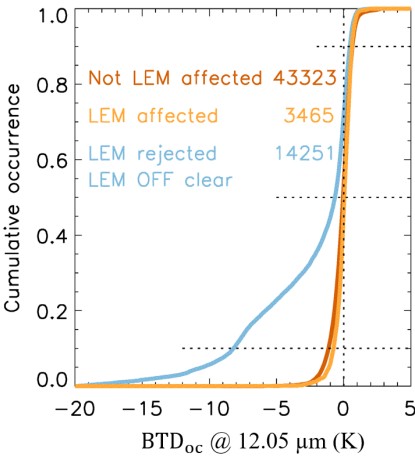

**Figure 12. Cumulative frequency of IIR 12.05 μm [observed − computed] brightness temperature differences in profiles classified as cloud-free by CALIOP over ocean for 10-20°S during January 2022 at night. LEM is enabled for profiles unaffected by LEM (red; LEM column QC = 0) and affected by LEM (yellow; LEM column QC = 1). LEM is disabled for profiles that would have been rejected by LEM (blue; any of LEM column QC bits 1–3 set). Horizontal dashed lines indicate 10, 50, and 90 % cumulative frequencies.**

The net impact of LEM on false feature elimination is demonstrated in Fig. 13 which compares the monthly daytime cloud amount with LEM enabled and LEM disabled. Cloud amount is calculated as the number of profiles containing any cloud detections at 333 m or 5 km horizontal resolution relative to all profiles observed during the month. No-confidence clouds are excluded (CAD < 20), consistent with standard quality filtering procedures in CALIOP products (Liu et al., 2019). The cloud amount is abnormally low in the SAA region when LEM is disabled (Fig. 13a). This is because cloud detections do not exist in the profile after no-confidence clouds are excluded. Enabling LEM ensures that only profiles with acceptable levels of low energy pulses contribute to the cloud amount calculation. Consequently, the cloud amount increases along the perimeter of the SAA (Fig. 13c), capturing a more realistic representation of the true cloud amount. For example, the high cloud amount along the western coast of Peru is expected due to the presence of low altitude marine clouds in this region (Dong and Minnis 2023). The cloud amount here was lower by ½ without LEM enabled. All profiles are rejected by LEM within the center of the SAA during this month due to excessive low energy pulses (Fig. 13b).





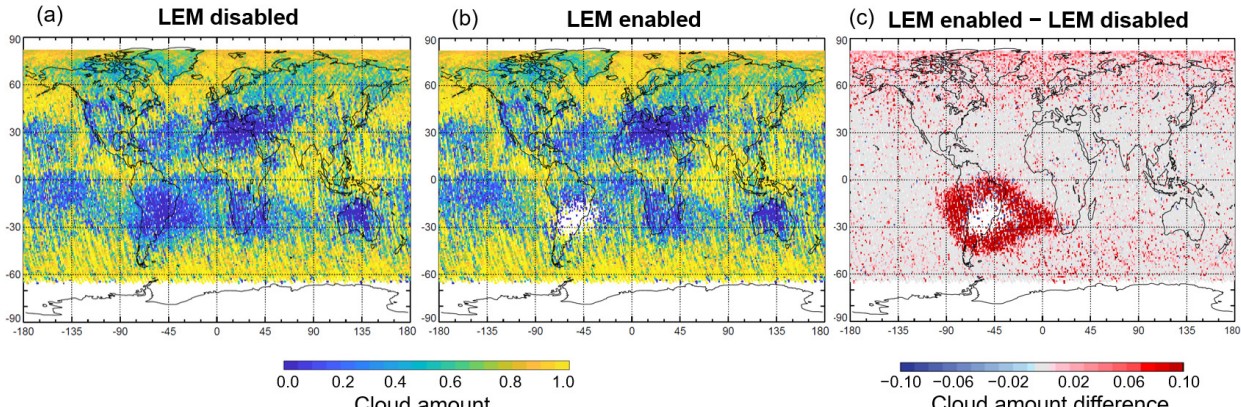

**Figure 13. Cloud amount fraction for July 2022, day, with (a) LEM disabled, (b) LEM enabled, and (c) the difference in cloud amount (enabled − disabled).**

LEM-affected profiles accepted for use in the level 2 analyses typically have reduced SNR due to intra-frame data rejection, and this reduction will affect feature detections. Table 4 summarizes the reduction in nighttime SNR for 5 km resolution LEM-affected profiles for 50°S–50°N, excluding the SAA during 2020–2021. Here, SNR is computed in clear-sky regions as the mean $\beta'_{532}$ divided by its standard deviation and multiplied by the square root of number of single-shot samples within each 5 km horizontal × 4 km vertical volume (i.e., $\mathrm{SNR} = \sqrt{N} \cdot \mu/\sigma$ ). Each LEM-affected profile is paired with the nearest

unaffected clear-sky profile along the lidar track, typically within 11 km (avg.), ensuring similar conditions are sampled. Mean SNR ($\mu_{SNR}$) is computed from all the LEM-affected and unaffected profiles separately and the relative difference is computed as:

$$\Delta\mathrm{SNR} = 100 \times \left(\overline{\mathrm{SNR}}_{\mathrm{affected}} / \overline{\mathrm{SNR}}_{\mathrm{unaffected}} - 1\right) \tag{8}$$

Most LEM-affected profiles contain ≤ 4 LEM-rejected single-shot profiles below 30 km (72 %), and thereby experience an SNR reduction of 6–9 %. A larger SNR reduction of 20–24 % occurs for the 23 % of LEM-affected profiles with 5–7 single-shot profiles rejected. Note that having this many single-shot profiles rejected in a 5 km profile outside the SAA was rare prior to 2020 (< 0.5 % of all profiles) and more common in 2022–2023 (~1 % of all profiles; Fig. S3). On occasions when 8 or more single-shot profiles are rejected (5% of LEM-affected profiles), the SNR reduction can reach 35

%. Prior to 2022, less than 0.5 % of all profiles experienced this magnitude of intra-frame LEM rejection, although this rate increased to 3–10% of all profiles in 2022 and 2023.





| | N rejected single-shots ≤ 4 (72 % of affected profiles) | | | N rejected single-shots 5–7 (23 % of affected profiles) | | | N rejected single-shots ≥ 8 (5% of affected profiles) | | |
|---|---|---|---|---|---|---|---|---|---|
| Altitude (km) | $\Delta$SNR (%) | $\Delta P_{fa}$ (%) | $\Delta R'_{thresh}$ (%) | $\Delta$SNR (%) | $\Delta P_{fa}$ (%) | $\Delta R'_{thresh}$ (%) | $\Delta$SNR (%) | $\Delta P_{fa}$ (%) | $\Delta R'_{thresh}$ (%) |
| 36 − 40 | −6.1 | +0.7 | +3.3 | −19.9 | +3.0 | +11.8 | −30.7 | +5.7 | +22.7 |
| 26 − 30 | −8.2 | +1.0 | +2.0 | −21.8 | +3.4 | +7.3 | −35.2 | +7.4 | +13.7 |
| 16 − 20 | −9.0 | +1.1 | +1.0 | −24.3 | +4.0 | +3.6 | −33.8 | +6.7 | +6.7 |
| 4 − 8 | −6.9 | +0.8 | +0.4 | −22.7 | +3.6 | +1.6 | −32.7 | +2.9 | +6.3 |

**Table 4. Percent relative difference in clear-sky nighttime SNR, absolute difference in probability of false detection ($P_{fa}$), and relative difference in attenuated scattering ratio threshold ($R'_{thresh}$) between LEM-affected profiles and LEM-unaffected profiles at 5 km horizontal resolution for the years 2020-2021, at latitudes ±50°N/S, excluding the SAA at night. Positive percentages indicate the LEM-affected value is larger.**


The probability of false detection increases in LEM-affected profiles due to the reduction in SNR. Based on straightforward detection theory (Kingston 1978), Eq. (9) estimates the minimum 532 nm scattering ratio required for detecting the top altitude of a particulate feature in a purely molecular atmosphere at night as a function of detection probability. A full derivation is given in Sect. 2.1 of Vaughan et al., 2005. It is cast in terms of molecular $SNR_m$. $N_f$ is the

excess noise factor of the CALIOP PMT detector, while $x_d$ and $x_{fa}$ are parameterized terms encapsulating the molecular and particulate signals and the detection threshold. They are related to the probabilities of true feature detection $P_d$ and false feature detection $P_{fa}$ by Eqs. (10–11). A value of $x_d = x_{fa} = 2.055$ yields a 98 % detection rate and a 2 % false detection rate. For this theoretical exercise, $x_d$ and $x_{fa}$ remain identical to simulate the optimum scenario where the feature detection threshold is selected at the lowest probability level where the molecular and particulate signal distributions overlap.


$$R_{min} = \left( \frac{x_d N_f}{2 SNR_m} + \sqrt{1 + \frac{x_{fa} N_f}{SNR_m} + \left( \frac{x_d N_f}{2 SNR_m} \right)^2} \right)^2 \qquad (9)$$

$$P_d(x_d) = 1 - \frac{1}{2} \operatorname{erfc}\left( \frac{x_d}{\sqrt{2}} \right) \qquad (10)$$

$$P_{fa}(x_{fa}) = \frac{1}{2} \operatorname{erfc}\left( \frac{x_{fa}}{\sqrt{2}} \right) \qquad (11)$$





For a constant value of $R_{\min}$, $x_d$ and $x_{\mathrm{fa}}$ reduce in proportion to the reduction in $\mathrm{SNR}_m$. Using the fractional change in SNR reported in Table 4 (not the percent), the increase in probability of false feature detection is computed for LEM-affected profiles by Eq. (12). For most 5 km resolution LEM-affected profiles with ≤ 4 rejected single-shot profiles, the theoretical false detection rate increases by ~1 %. A higher increase of 3–4 % is postulated for profiles with 5–7 single-shot rejections, and an increase upwards of 7% for the 5% of affected profiles with 8 or more single-shot rejections.


$$\Delta P_{\mathrm{fa}} = 100 \times [P_{\mathrm{fa}}(\Delta \mathrm{SNR} \cdot x_{\mathrm{fa}}) - P_{\mathrm{fa}}(x_{\mathrm{fa}})] \tag{12}$$

Note however that the feature detection threshold actually used in the CALIOP level 2 algorithm, $R'_{\mathrm{thresh}}$, increases to somewhat limit the possibility of false detections in LEM-affected profiles. The threshold depends, in part, on the measured backscatter variance (MBV) which characterizes detector dark current and solar background noise (Vaughan et al.,

2005). It is computed from the RMS noise, in digitizer counts, measured in a signal-free region of the atmosphere (65–80 km). During the MBV conversion from units of counts to units of backscatter (km$^{-1}$sr$^{-1}$), 5 km horizontal averages are divided by the mean laser energy, among other terms. Because the mean laser energy includes low energy pulses, it will be biased low in LEM-affected profiles. This increases the MBV and therefore $R'_{\mathrm{thresh}}$ as well. Table 4 shows the relative increase in $R'_{\mathrm{thresh}}$ for LEM-affected profiles is 0.4–3.4 %, depending on altitude, for profiles with ≤ 4 single-shot rejections.

Raising the feature detection threshold helps to counteract the influence of enhanced noise. The increased probability of false detections is more relevant in low signal scenarios, for example with weakly backscattering layers or beneath substantial attenuation. Stronger signal scenarios will be more resilient to false feature detection.

**6.2 Impacts on detected layer properties**

Despite the reduction in SNR and small increase in probability of false-detections, the spatial and optical properties of LEM-

affected features are similar to those of layers that are wholly unaffected by low energy pulses. Figure 14 shows the distribution of key properties used for feature classification and subtyping for layers detected in 2021 at night at 5 km horizontal resolution over 50°N–50°S, excluding the SAA. Affected layers are identified using the LEM feature QC flag. To ensure comparable layers are sampled from the same geographical region, the 10 nearest unaffected layers having the same feature type and horizontal averaging required for detection are sampled for each affected layer. On average, affected and

unaffected layers are within 230 km of each other.





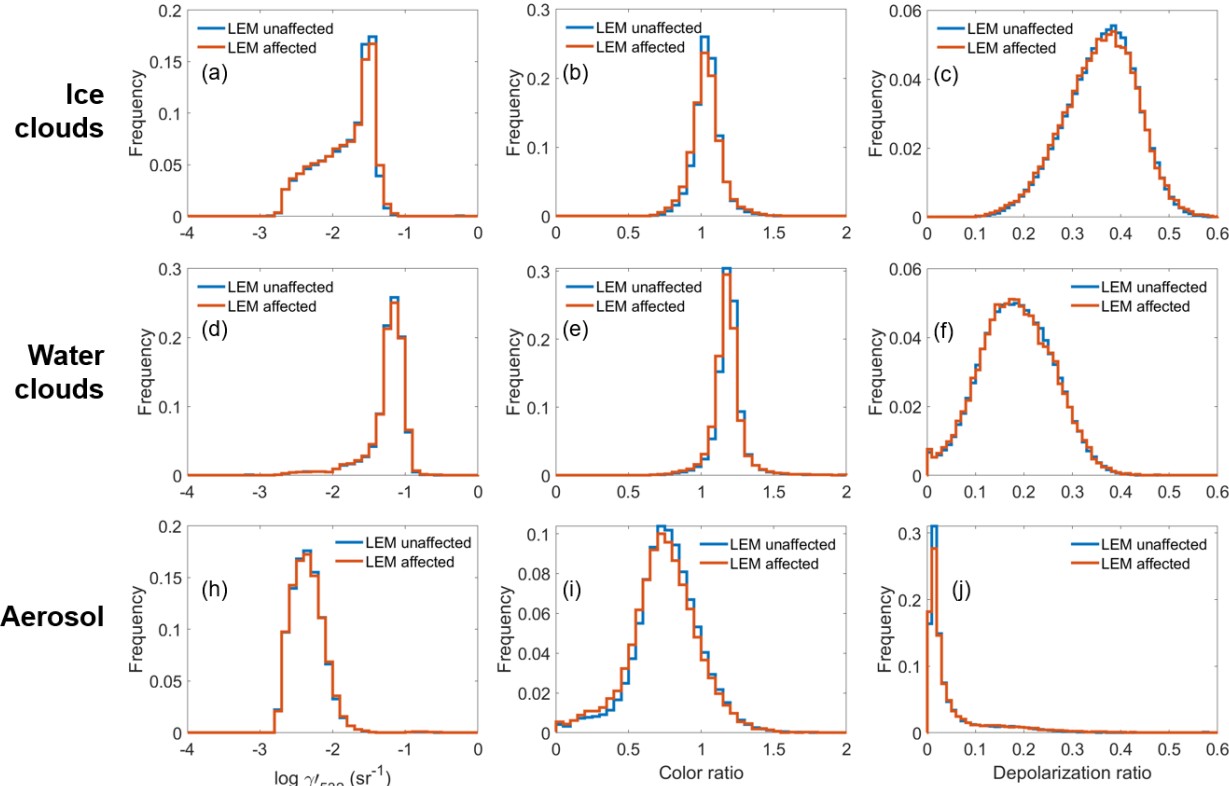

**Figure 14. Frequency distributions of layer-integrated 532 nm attenuated backscatter ($\gamma'_{532}$), attenuated backscatter color ratio ($\chi'$), and volume depolarization ratio ($\delta_v$) for ice clouds (a-c), water clouds (d-f), and aerosol layers (h-j) detected at 5 km horizontal resolution between 50°S-50°N, excluding the SAA, during 2021 at night. LEM-affected layers are identified by LEM feature QC flag = 1.**

There is reasonable overlap in the distributions of unaffected and affected layers for most layer properties, though unequal variance t-tests conclude that there are statistically significant differences at the 5 % confidence level. Attenuated backscatter color ratios ($\chi'$) of affected layers appear slightly lower compared to unaffected layers. This is apparent in Table 5, where the first three columns compare the median and median absolute deviation (MAD) of these distributions. For most layer properties, median relative difference magnitudes are less than 1 % for affected layers except for ice cloud integrated attenuated backscatter ($\gamma'_{532}$) and aerosol $\chi'$ and $\delta_v$ where relative differences are < 6 %. Based on the MAD relative differences, the affected layer property distributions are mostly broader by 1–14% compared to unaffected distributions. The final column of Table 5 characterizes the relative differences between all pairings of affected and unaffected layers, where each relative difference is computed between the affected layer and the median of the 10 nearest unaffected layers. The median of the relative differences is less than 1 % for all feature types and layer properties, with aerosol $\chi'$ being the exception at −2.9 %. These levels of differences for affected layers compared to unaffected layers will have small effects on CALIOP feature classification and subtyping algorithms on average, because thresholds for these algorithms are spaced at





coarser intervals. For example, estimated particulate depolarization ratio thresholds to discriminate smoke from dust are less than 0.075 and greater than 0.20, respectively. A 0.3 % relative difference in $\delta_v$ is small in comparison. Analogous histograms and statistics as Fig. 14 and Table 5 are given for daytime observations in Fig. S4 and Table S1. Relative differences between paired observations are slightly larger during the day, at 1–3 % for most layer properties. Aerosol $\delta_v$ is the exception at 6.8 % larger for affected layers compared to unaffected layers.


| | | | Median ± MAD | Relative difference in median | Relative difference in MAD | Median ± MAD of all relative differences |
|---|---|---|---|---|---|---|
| Ice clouds | $\gamma'_{532}$ | Unaffected | 0.017 ± 0.010 | −2.88 % | 5.63 % | 0.33 ± 49.9 % |
| | | Affected | 0.016 ± 0.011 | | | |
| | $\chi'$ | Unaffected | 1.015 ± 0.072 | −0.60 % | 12.2 % | −0.59 ± 7.77 % |
| | | Affected | 1.009 ± 0.080 | | | |
| | $\delta_v$ | Unaffected | 0.360 ± 0.059 | −0.54 % | 2.90 % | −0.23 ± 12.1 % |
| | | Affected | 0.358 ± 0.061 | | | |
| Water clouds | $\gamma'_{532}$ | Unaffected | 0.058 ± 0.031 | −0.11 % | −30.9 % | −0.13 ± 34.1 % |
| | | Affected | 0.058 ± 0.021 | | | |
| | $\chi'$ | Unaffected | 1.164 ± 0.091 | −0.80 % | 11.3 % | −0.76 ± 8.21 % |
| | | Affected | 1.155 ± 0.101 | | | |
| | $\delta_v$ | Unaffected | 0.179 ± 0.060 | −0.79 % | 0.64 % | −0.39 ± 25.1 % |
| | | Affected | 0.178 ± 0.061 | | | |
| Aerosol | $\gamma'_{532}$ | Unaffected | 0.004 ± 0.003 | 0.30 % | 10.3 % | −0.05 ± 58.9 % |
| | | Affected | 0.004 ± 0.003 | | | |
| | $\chi'$ | Unaffected | 0.741 ± 0.175 | −2.69 % | 5.20 % | −2.87 ± 26.1 % |
| | | Affected | 0.721 ± 0.184 | | | |
| | $\delta_v$ | Unaffected | 0.016 ± 0.046 | 5.08 % | 13.8 % | 0.27 ± 167 % |
| | | Affected | 0.017 ± 0.053 | | | |

**Table 5. Median and median absolute deviation (MAD) of layer-integrated 532 nm attenuated backscatter ($\gamma'_{532}$), attenuated backscatter color ratio ($\chi'$), and volume depolarization ratio ($\delta_v$) for ice clouds water cloud, and aerosol layers detected at 5 km horizontal resolution between 50°S–50°N, excluding the SAA, during 2021 at night. LEM-affected layers are identified by LEM feature QC flag = 1. The final two columns give the median ± MAD of all the pairs of affected and unaffected layers.**


Feature classification confidence is also similar for LEM-affected and unaffected layers. Table 6 compares confidence levels for clouds and aerosols at three horizontal averaging resolutions. Affected and unaffected layers are paired



using the method described in the previous paragraph, except here cloud layers of all ice-water phase determinations are combined and there are 5 unaffected layers sampled for each 20 km resolution affected layer and 2 sampled for each 80 km

resolution affected layer, respectively. The occurrence of high confidence type classifications is similar, having differences within 0.4–2.5 % for unaffected and affected layers. No confidence classifications are also similar, having differences within 0.2–0.8 %, depending on the horizontal averaging and feature type.

|  |  | N layers | No confidence | Low confidence | Medium confidence | High confidence |
|---|---|---|---|---|---|---|
| Clouds | Unaffected by LEM | 668060 | 4.0 % | 1.8 % | 1.8 % | 92.5 % |
| 5 km | Affected by LEM | 66806 | 4.6 % | 2.0 % | 2.0 % | 91.3 % |
| Clouds | Unaffected by LEM | 115485 | 48.9 % | 22.8 % | 3.9 % | 24.4 % |
| 20 km | Affected by LEM | 23097 | 49.7 % | 23.4 % | 3.7 % | 23.2 % |
| Clouds | Unaffected by LEM | 16248 | 29.5 % | 42.9 % | 4.3 % | 23.3 % |
| 80 km | Affected by LEM | 8124 | 28.7 % | 43.3 % | 4.3 % | 23.8 % |
| Aerosol | Unaffected by LEM | 176520 | 2.1 % | 3.0 % | 2.5 % | 93.0 % |
| 5 km | Affected by LEM | 17652 | 2.9 % | 3.4 % | 3.0 % | 90.5 % |
| Aerosol | Unaffected by LEM | 157450 | 11.4 % | 4.4 % | 3.8 % | 80.0 % |
| 20 km | Affected by LEM | 31490 | 11.6 % | 4.7 % | 4.1 % | 79.6 % |
| Aerosol | Unaffected by LEM | 43464 | 8.6 % | 3.3 % | 2.9 % | 85.3 % |
| 80 km | Affected by LEM | 21732 | 9.0 % | 3.5 % | 2.9 % | 84.6 % |

**Table 6. Feature classification confidence for layers that are unaffected and affected by LEM between 50°S-50°N, excluding the SAA, during 2021 at night. No confidence: |CAD score| < 20; low confidence: 20 ≤ |CAD score| < 50; medium confidence: 50 ≤ |CAD score| < 70; high confidence: |CAD score| ≥ 70.**

### 6.3 Impacts on extinction retrievals

The final step in level 2 data processing after features are detected and classified is the retrieval of particulate extinction and

backscatter coefficients. Typically, the extinction retrieval process is successful, with failures occurring the minority of the time (6–7% of cloud solutions and 3 % of aerosol solutions). To evaluate the potential influence of LEM on the retrieval process, Fig. 15 compares the frequency of suspicious and failed 532 nm extinction retrievals for the unaffected and affected layers paired in the previous paragraph. This information is contained within the level 2 extinction QC flag SDS, a bit-mapped quantity that summarizes the final disposition of the extinction retrieval for each layer (Young et al., 2018). The

interpretation of the three most common "bad" bits is denoted. LEM-affected cloud and aerosol layers experience increases of 1.2 % and 0.2 % in retrievals where lidar ratio adjustments cannot yield successful solutions (primarily bits 2 and 8),





respectively. Despite this increase, the majority of extinction solutions are successful for affected layers (92.7 % for cloud and 97.0 % for aerosol), suggesting that extinction retrieval behavior is nominal in level 2 data products.

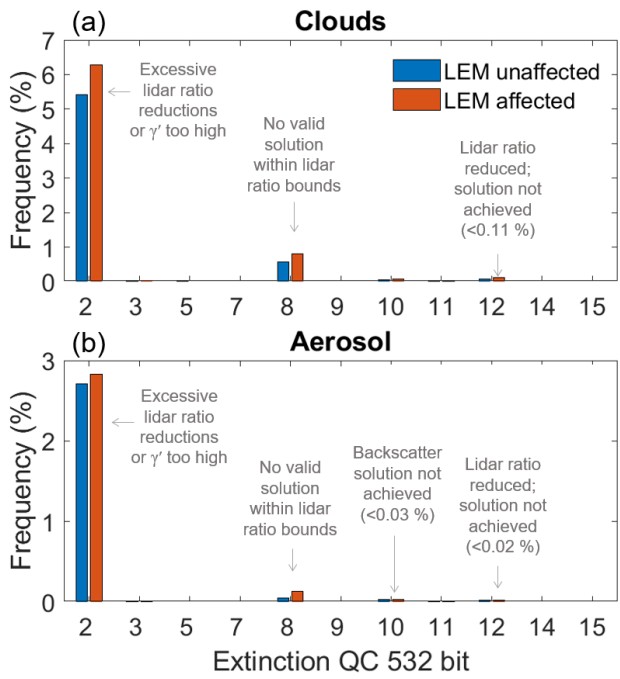


**Figure 15. Frequency of extinction QC flag bits (0-based indexing) indicating suspicious or failed retrievals out of all (a) cloud and (b) aerosol layers detected between 50°S-50°N, excluding the SAA, during 2021. LEM-affected layers are identified by LEM feature QC flag = 1.**

## 7 LEM adaptations in level 3 products

The CALIOP data record includes a suite of level 3 products which are monthly, quality-screened aggregates of level 1B and level 2 data. These products are updated in the final CALIOP data release in 2025 to include the benefits of level 2 LEM filtering. The implementation strategy of LEM in level 3 varies depending on the focus of the product, providing useful examples of how LEM-affected data and LEM quality flags can be used for different applications. A full analysis on the impacts of LEM filtering on these products will be included in data quality summaries that will accompany each level 3 data

release. This section provides a brief overview of LEM adaptations planned for the level 3 products.

The level 3 tropospheric aerosol products are aggregates of level 2 aerosol extinction for the lower troposphere, reported separately by cloud cover (Tackett et al., 2018). Given that the behavior of level 2 extinction retrievals is similar for LEM-affected and unaffected layers with marginal increases in uncertainty, the final release of this level 3 product will allow LEM-affected profiles to contribute to monthly averages. Profiles rejected by LEM will be identified using the LEM column

QC flag and excluded from aggregation.



The level 3 ice cloud product summarizes level 2 extinction retrievals and ice water content for ice clouds (Winker et al., 2024). Similar to the level 3 tropospheric aerosol product, 5 km profiles rejected by LEM will be excluded from aggregation and LEM-affected profiles will be allowed to contribute. The level 3 cloud occurrence and level 3 GEWEX cloud products report cloud detections based on single-shot and 5 km resolution observations. To ensure only full profiles are evaluated when determining cloud presence, entire single-shot profiles will be excluded when any part of the profile is rejected by LEM. All three level 3 cloud products will be modified to exclude profiles where LEM prevents feature detection attempts at 20 or 80 km resolution (identified by bits 4 and 5 of the LEM column QC flag). This is to avoid counting profiles as cloud-free where features are not sought due to LEM rules.

The level 3 stratospheric aerosol product reports aerosol extinction profiles for the stratosphere retrieved from aggregates of cloud-cleared level 1B attenuated backscatter (Kar et al., 2019). Features reported in level 2 products are used to remove clouds. Because the background stratospheric aerosol signal is small, it must be aggressive with cloud-clearing to avoid biases caused by cloud contamination. For this reason, the updated product will adopt a more conservative approach to LEM. All level 1B single-shot profiles within a 5 km frame will be excluded if any single-shot profile contains low energy pulses based on the level 2 LEM column QC flag. Further, profiles in 20 and 80 km data segments will be excluded where LEM prohibits feature detection attempts at these resolutions to avoid undetected clouds from contributing to the average. Non-zero values of the LEM column QC flag will be used to identify affected data.

## 8 Summary

Overcoming the impact of low energy laser pulses on data quality is important to maintain the integrity of the CALIOP observational record during the latter years of the mission. The low laser energy laser pulses were associated with coronal discharge and radiation-induced arcing within the laser canister as internal pressure levels decreased. Beginning in mid-2017, this caused increasing occurrences of intermittent low energy pulses that affected observations predominantly over the South Atlantic Anomaly (SAA) region. Outside the SAA, low energy pulses occurred at lesser, though non-negligible rates, approaching 2–8 % of all pulses during 2023. In response to the increasing degradation in data quality, the CALIOP mission developed a suite of low energy mitigation (LEM) procedures for level 1–3 data products that apply data rejection on small, targeted scales to minimize the effects of irretrievable data loss. LEM accomplishes this by rejecting profiles with laser energies below required thresholds and by accounting for the altitude-varying averaging scheme applied onboard the satellite. Horizontally averaged profiles that contain excessive numbers of low energy pulses are rejected, while altitude regions containing permissible numbers of low energy pulses or single-shot LEM rejection are allowed to remain; the latter profiles are considered "affected" by LEM. New data quality flags are added to the CALIOP level 2 data products to clearly indicate where and why data is rejected and/or affected. LEM is implemented in the final version 5 CALIOP data release in 2025.





The analysis presented demonstrates that LEM effectively rejects data affected by low energy pulses that would otherwise degrade data quality. Calibration biases are corrected, and calibration uncertainties are reduced for 532 nm daytime and 1064 nm attenuated backscatter coefficients at SAA-latitudes. A revised level 1 energy-normalization procedure compensates for low energy pulses, improving the SNR at altitudes where onboard horizontal averaging is applied. LEM rejection in V5 level 2 data products reduces false features that were detected in V4.51 and finds more true features that were previously classified as clear sky due to poor SNR. This leads to a more accurate representation of the real atmospheric state than if LEM were not applied. The SNR in frames (5 km horizontal averages) that are affected by LEM, but not rejected, is reduced in proportion to the number of low energy pulses within their extent. Most frames with a smaller number of low energy pulses (73 %) experience an SNR reduction of 6–9 %, while the minority with a larger number (28 %) experience an SNR reduction of 22–26 %. This increases the probability of false feature detections in LEM-affected frames by 1 % and 4 % for these cases, respectively. However, a small increase in the CALIOP feature detection threshold occurs in these frames, which reduces the chances of false detection to some extent. Features affected by LEM are similar to unaffected features, in terms of measured layer-averaged properties that are important for classification and subtyping: namely, integrated attenuated backscatter at 532 nm, volume depolarization ratio, and attenuated total backscatter color ratio. Distributions of these properties and their average values are similar for LEM-affected and unaffected ice clouds, water clouds, and aerosols. Cloud-aerosol discrimination confidence scores indicate that high-confidence feature classification occurrences are similar to within < 2.5 %. The occurrence of extinction retrieval failures also increases by at most 1.2 % for LEM-affected features relative to unaffected features, though the majority of retrievals remain successful (> 93 %).

Taken together, this evidence suggests that the major impacts of low energy pulses on data quality have been mitigated by rejecting profiles containing too many low energy laser pulses. Most of the remaining features that are LEM-affected experience minor impacts due to reduced SNR. Small increases in extinction retrieval errors are simple to reject with standard quality filtering procedures recommended by the CALIPSO mission (Liu et al., 2019; Tackett et al., 2018; Winker et al., 2024). The new LEM quality control flags allow easy identification of LEM-affected layers, giving data users control over how these features should be used in their analyses. The discussion on adaptations of LEM for CALIOP level 3 products provides examples of how the flags are applied for several use-cases. Though data loss was inevitable, particularly over the SAA region, the adoption of LEM in the final version 5 data release has substantially improved the integrity of the long-term record provided by CALIOP.

As a concluding remark, the LEM algorithm is more complicated than it ideally should be. This complication is due to the altitude-varying horizontal averaging that is performed onboard the satellite to reduce the volume of downlinked data, which irrevocably propagates the malign effects of low energy pulses into horizontally-adjacent range bins. Because the horizontal averaging scales are different within each altitude region, multiple LEM acceptance requirements are needed to identify suspicious data. This has led to an admittedly convoluted solution with intertwined requirements. By contrast, if the CALIOP data had been averaged onboard at a uniform horizontal scale for all altitudes, the LEM algorithm would have been much simpler: just reject any profile with too many low energy pulses and ensure the resulting SNR is acceptable. Over the




past 17 years CALIOP algorithm developers have found that altitude-varying onboard horizontal averaging caused onerous levels of complexity in many situations. Uniform horizontal averaging onboard, or no horizontal averaging at all, is recommended for all future spaceborne lidar missions.

**Appendix A: Coordinates used for the South Atlantic Anomaly (SAA)**

The following coordinates define a polygon that encapsulates the SAA, based on the observed locations of frequent low energy laser pulses. Given that the size of the SAA expands and contracts depending on the level of solar activity (Noel et al., 2014), the polygon is intentionally broad and will be larger than the actual SAA at any given time.

| Latitude (° N) | Longitude (° E) | Latitude (° N) | Longitude (°E) |
|---|---|---|---|
| −10 | −94 | −29 | 20 |
| 0 | −74 | −31 | 4 |
| 6 | −50 | −34 | −8 |
| 7 | −30 | −47 | −40 |
| 2 | −18 | −47 | −71 |
| −10 | 0 | −39 | −84 |
| −18 | 20 | −27 | −91 |
| −25 | 26 | −10 | −94 |

**Table A1 Latitude and longitude coordinates for the SAA.**

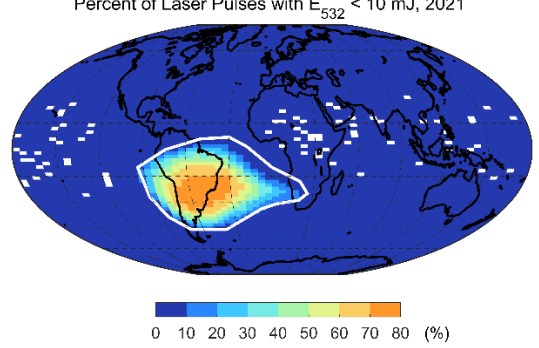

**Figure A1 Polygon defining the SAA (white outline) and frequency of laser pulses with 532 nm energy < 10 mJ during 2021.**





**Appendix B: Bit assignments for LEM QC flags**

The following tables define the bit assignments and interpretations of the new bit-mapped LEM QC flags added to the V5 level 2 data products. The Low Energy Mitigation Column QC Flag is reported at single-shot, 1 km, and 5 km resolution, 690 depending on the data product. The VFM Feature Detection Quality Flag is reported in the VFM product and indicates for each 5/3 km data segment which single-shot range bins contain low energy, which are rejected by LEM, and which contributed to feature detection at the indicated resolution.

| Bit | Interpretation |
|---|---|
| 0 | Profile contains LEM-affected data (data has been rejected or contains low energy pulses that LEM accepts) |
| 1 | Profile belongs to a frame that has been rejected due to too many unusable profiles |
| 2 | Profile belongs to a frame that has been rejected due to too many rejected subregions in altitude region 3 |
| 3 | Profile belongs to a frame that has been rejected due to too many rejected subregions in altitude region 4 |
| 4 | Feature detection not performed at 20 km resolution in this profile due to too many rejected frames |
| 5 | Feature detection not performed at 80 km resolution in this profile due to too many rejected frames |
| 6 | Unused |
| 7 | Profile has data rejected in altitude regions 1 and 2 (only reported at single-shot resolution) |
| 8 | Profile has data rejected in altitude region 3 (only reported at single-shot resolution) |
| 9 | Profile has data rejected in altitude region 4 (only reported at single-shot resolution) |
| 10 | Profile does not have low energy, but data is rejected in regions 1 & 2 due to rejected data in altitude region 3 (only reported at single-shot resolution) |

**Table B1. Bit assignments and interpretations for the Low Energy Mitigation Column QC Flag.**

| Bit | Interpretation |
|---|---|
| 0 | First single-shot profile has low energy |
| 1 | Second single-shot profile has low energy |
| 2 | Third single-shot profile has low energy |
| 3 | Fourth single-shot profile has low energy |
| 4 | Fifth single-shot profile has low energy |
| 5 | Bin in first single-shot profile is rejected by LEM |
| 6 | Bin in second single-shot profile is rejected by LEM |
| 7 | Bin in third single-shot profile is rejected by LEM |
| 8 | Bin in fourth single-shot profile is rejected by LEM |



| 9 | Bin in fifth single-shot profile is rejected by LEM |
|---|---|
| 10 | Contributed to feature detection at 1/3 km resolution |
| 11 | Contributed to feature detection at 1 km resolution |
| 12 | Contributed to feature detection at 5 km resolution |
| 13 | Contributed to feature detection at 20 km resolution |
| 14 | Contributed to feature detection at 80 km resolution |

**Table B2. Bit assignments and interpretations for the VFM Feature Detection Quality Flag**

| Value | Interpretation |
|---|---|
| 0 | Layer contains no low energy pulses or regions rejected by LEM |
| 1 | Some portion of the layer contains low energy pulses and/or data rejected by LEM |

**Table B3. Bit assignments and interpretations for the Low Energy Mitigation Feature QC Flag.**

**Data availability**

The CALIOP and IIR data used in this analysis are available through the NASA Langley Research Center Atmospheric Science Data Center (ASDC), https://asdc.larc.nasa.gov/ (last access: 9 May 2025): V4.1 CALIOP level 1B (NASA/LARC/SD/ASDC 2016b), V4.51 CALIOP level 1B (NASA/LARC/SD/ASDC 2022a), V4.5 CALIOP level 2 merged layer (NASA/LARC/SD/ASDC 2023a), V4.5 CALIOP level 2 vertical feature mask (NASA/LARC/SD/ASDC 2023b), V4.51 IIR level 2 track (NASA/LARC/SD/ASDC 2023c). Version 5.0 CALIOP level 1B, level 2 merged layer, and

vertical feature mask products will be publicly available at the ASDC in mid-2025. << NOTE TO EDITOR: CITATIONS FOR V5.0 WILL BE ADDED TO THIS SENTENCE PRIOR TO FINAL SUBMISSION IF AVAILABLE.

**Author contributions**

The CALIOP low energy mitigation (LEM) algorithm was conceived by JT, RR, AG, XC, BG, and MV. Analyses and visualizations were prepared by JT except for Fig. 2 (RV), Table 3 (MV), Fig. 12 (AG), and Fig. 13 (XC). The manuscript

was written by JT with extensive contributions from RR, AG, MV, DW, CT, and JK; technical oversight provided by RV, K-PL, and BG. Implementation of LEM into CALIOP production code was conducted by RR, K-PL, and BG.

**Competing interests**

The authors declare that they have no conflict of interest.



## Acknowledgements

The authors are indebted to Bill Hunt, Floyd Hovis, and Carl Weimer for their analysis on the low energy pulse phenomenon and the 2009 laser switch operation. Ken Beaumont and Tim Murray of the CALIPSO Data Management Team are also acknowledged for their assistance in generating test data sets used during preliminary manuscript preparations.

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
