# Peer review of "Mitigating impacts of low energy laser pulses on CALIOP data products"

_EGUsphere, 2025_

## Author Comment (AC1)

*This manuscript presents a well-structured study on the development and implementation of low-energy mitigation (LEM) procedures to address the impacts of low-energy laser pulses on CALIOP observations during the latter years of the CALIPSO mission. The results show that LEM effectively improves data quality, corrects calibration biases, and provides new quality control tools that will greatly enrich CALIOP data products during the final seven years of low-energy operation. I recommend this work for publication in Atmospheric Measurement Techniques (AMT).*

We thank the referee for their review of the manuscript.

---

## Author Comment (AC2)

*This manuscript presents a suite of low energy mitigation procedures to reject data that is impacted by the low energy pulses. This deployment of the procedure will minimize the data loss and ensure the qualities of the aerosol data products from CALIOP. The benefits of this study would be great for the enriching of CALIOP data products during the period of low energy operation in the final seven years of CALIPSO mission. The manuscript is well written and provides very detailed information. Hence, I recommend the acceptance of this manuscript after the necessary revisions.*

We thank the referee for their thorough evaluation of the manuscript.

The general comments are listed below:

1. Could the authors provide some explanations regarding the different behaviors of the laser energy changing inside/outside SAA? Only because of the solar particle flux? Even though authors state that the mechanisms responsible for the observed variability in the frequency of low energy pulses may never be fully understood. But this is really meaningful, considering the lessons learnt from this mission will surely support the maintain/operation of ATLID/EarthCARE and ACDL/DQ-1.

    This is a good suggestion, and one that is difficult to answer with certainty. As demonstrated by Rodriguez et al., 2022, it is quite likely that solar particle flux played a role in the behavior of the low energy pulse time series. However, they concluded that the solar particle flux alone was not enough to explain the trend in low energy pulses. This implies some physical mechanism(s) inside the laser canister or associated electronics were involved. It is not possible to know the exact mechanisms without manual inspection of the instrument, leaving all mechanisms considered by mission analysts to have insufficient evidence to be published at this time. However, comment #1 makes a good suggestion to include more information about the relevance of solar particle flux inside/outside the SAA. The Rodriguez et al., 2022 study is the key source for this analysis, so we have added the following synopsis to the end of the introduction (lines ~151-163).

    "To determine the impacts of the space radiation environment on low energy pulse behavior, Rodriguez et al., 2022 compared the occurrence frequency and magnitude of CALIOP low energy pulses with observations of charged particle fluxes from the NOAA-19 Space Environment Monitor. The study found a clear relationship between low energy laser pulses within the SAA to highly energetic protons from the inner Van Allen belt, and that higher proton flux levels were spatially associated with larger reductions in pulse energy. Laser pulses having a small reduction of < 5 mJ from nominal were found to be more closely associated with less energic electrons in the outer Van Allen belt and galactic cosmic rays during the 2017 timeframe. However, the study found that these less energetic particles

were associated with laser pulse energy reductions of all magnitudes by 2021, including near-zero energy pulses, suggesting the impact of the space radiation environment on the behavior of low energy pulses had changed. Ultimately, Rodriguez et al., 2022 concluded that variations in the space radiation environment could not alone explain the behavior of the low energy time series. This implies physical mechanisms inside the laser canister or electronics were also partly responsible. Since the instrument cannot be manually inspected, the mechanisms responsible for the observed variability in the frequency of low energy pulses may never be fully understood."

2. *There are too many sections in this manuscript. I propose the authors to reorganize the structure of this manuscript.*

Thank you for the suggestion. The number of sections in the manuscript is intentionally designed to help readers of different interests to find information quickly. For example, readers who are not interested in the details of the LEM algorithm and just want to understand what LEM-rejected data looks like in the data products can find this in *Sect. 4.3 How LEM-rejected data appears in level 2 products*. Likewise, if this reader just wants to understand the impacts of LEM on feature detection, they can find this information in *Sect. 6.1*. We believe that the manuscript is intuitively organized in its current form and the subsections help readers navigate the (admittedly lengthy) manuscript more easily. The current structure also ensures each topic is placed in a coherent and digestible subsection. For these reasons, we prefer to keep the current organization of the manuscript.

3. *In section 4.2, is it possible to prepare a flowchart to describe the procedure of LEM, with remarking the difference between the LEM and traditional method?*

The LEM requirements described in section 4.2 and Table 2 are applied simultaneously in a logical OR statement so there is not a specific ordering that could be illustrated by a flow chart. However, comment #3 makes a good point that this was not clear in the original manuscript. To clarify, we added the following sentence to lines ~277-279:

"In the software implementation of these rules, all requirements are evaluated simultaneously using a logical OR statement, such that the relevant data is rejected if any subregion or frame acceptance requirements are not met in the given profile."

*The technical corrections:*

1. *Line 40, "…during the last six years…….", however, the duration is 7 years in the description of the abstract. Please make them consistent.*

Thank you for catching this inconsistency. Lines 39-40 have been corrected as below. Also, line 645 in the summary now indicates the low energy issue began in mid-2016 instead of mid-2017.

"However, the greatest data quality challenge experienced by CALIOP was an increasing number of intermittent low energy laser pulses from the lidar transmitter during the last seven years of the mission (mid-2016 to 2023)."

2. *Line 168/174, the full name of LEM is not necessary anymore. The abbreviation has been described in the previous text. The authors should check/address this issue throughout the manuscript.*

The re-definition of LEM has been removed from line 168. We prefer to keep the definition of LEM at line 174 because this is the beginning of the section that formally describes the LEM algorithm. This allows readers who decide to beginning reading at section 4 to understand the definition. We scanned the remainder of the manuscript and ensured no other re-definitions exists.

3. *Figure 6, I propose the authors add the "Region 1" in this figure as well.*

Added as suggested.

4. *Figure 8, the titles of "day calibration" should be changed to "daytime calibration".*

Changed as suggested.